# GenLoco: Generalized Locomotion Controllers for Quadrupedal Robots

**Gilbert Feng**[1*]**, Hongbo Zhang**[2*]**, Zhongyu Li**[1]**, Xue Bin Peng**[1]**, Bhuvan Basireddy**[1]**,**
**Linzhu Yue**[2]**, Zhitao Song**[2]**, Lizhi Yang**[1]**, Yunhui Liu**[2]**, Koushil Sreenath**[1]**, Sergey Levine**[1]
[1] University of California, Berkeley    [2] The Chinese University of Hong Kong

**Abstract:** Recent years have seen a surge in commercially-available and affordable quadrupedal robots, with many of these platforms being actively used in research and industry. As the availability of legged robots grows, so does the need for controllers that enable these robots to perform useful skills. However, most learning-based frameworks for controller development focus on training robot-specific controllers, a process that needs to be repeated for every new robot. In this work, we introduce a framework for training generalized locomotion (GenLoco) controllers for quadrupedal robots. Our framework synthesizes general-purpose locomotion controllers that can be deployed on a large variety of quadrupedal robots with similar morphologies. We present a simple but effective morphology randomization method that procedurally generates a diverse set of simulated robots for training. We show that by training a controller on this large set of simulated robots, our models acquire more general control strategies that can be directly transferred to novel simulated and real-world robots with diverse morphologies, which were not observed during training. (Code and pretrained policies: `https://github.com/HybridRobotics/GenLoco`, Video: `https://youtu.be/5QUs32MjNu4`)

**Keywords:** Legged Locomotion, Reinforcement Learning, Transfer Learning

## 1 Introduction

Just as more general-purpose models have gained prominence in supervised learning domains, with broadly applicable language models that can solve a variety of NLP tasks [1, 2] and language-conditioned visual recognition or image generation models that can be applied to a variety of problems and settings [3, 4], so too we might expect that more powerful robotic learning systems might enable more broadly applicable robotic controllers. In particular, in settings where a variety of robotic platforms share the same basic morphology (e.g., the set of commonly used quadrupedal bodies, the set of 7 DoF robotic arms, the set of quadcopters, etc.), we might expect that it should be possible to train control policies that can be applied broadly to all robots within a particular set. If this were possible, then practitioners who want to make use of learned policies would not need to start by training policies of their own for their own robot, but could instead simply download a pre-trained policy for the general robot class from the web, and then deploy directly on their platform. In this paper, we take a step toward this vision in the particular setting of quadrupedal locomotion.

Quadrupedal locomotion offers the potential for robotic agents to traverse and operate in complex unstructured environments. However, designing effective locomotion controllers for quadrupedal robots is challenging, as it typically requires detailed knowledge of the dynamics of a particular system and careful controller design for each desired skill. Model-free reinforcement learning (RL) provides a paradigm that can automate much of the controller engineering process, where an agent learns locomotion skills automatically through trial-and-error. RL techniques have been effective for developing locomotion controllers for a large variety of quadrupedal robots [5, 6, 7, 8, 9, 10, 11, 12, 13]. While RL provides a general framework that can in principle be applied to any robot, the resulting controllers are most often specific to the particular robot that they were trained on. Therefore, these controllers will generally be ineffective when deployed on another robot, and new

---

* equal contribution

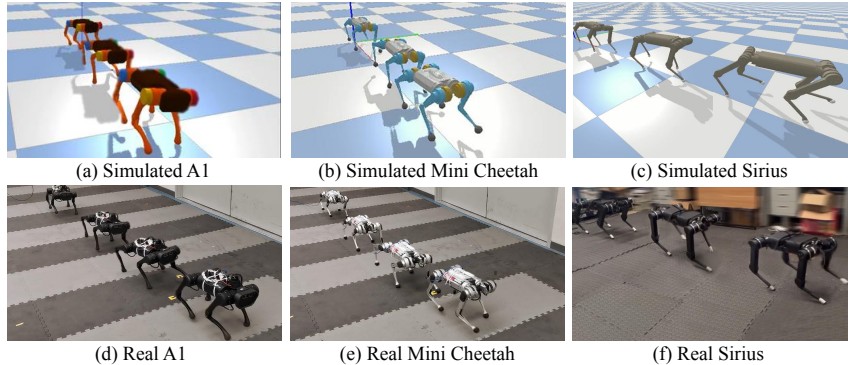

| (a) Simulated A1 | (b) Simulated Mini Cheetah | (c) Simulated Sirius |
| (d) Real A1 | (e) Real Mini Cheetah | (f) Real Sirius |

Figure 1: Testing of different simulated and real quadrupedal robots (A1, Mini Cheetah and Sirius) performing pacing gaits using a single locomotion controller. Our controllers can be directly deployed from simulation to the real world and across robots with different morphologies (e.g., body size, leg lengths, masses, etc.) and dynamics, without explicitly training on the specific robots used during testing.

controllers will need to be trained from scratch specifically for the new system. This requires either extensive data collection or, if using simulation to real-world transfer, detailed simulated models.

Recent years have seen the emergence of a growing catalog of commercially available quadrupedal robots, with many of these systems converging on similar body plans. This similarity in the morphologies of quadrupedal robots may lead one to wonder: is it possible to create a generalized locomotion controller that can be broadly applied to different quadrupedal robots? If so, such generalized controllers can greatly reduce the labor-intensive process of continually developing robot-specific controllers for new systems. In this work, we present a framework for developing more general locomotion controllers, which can be applied to a range of different quadrupedal robots. We focus our investigation on the setting where the test-time morphology is unknown, and thus the goal is to develop controllers that generalize to new robots not observed during training. To this end, we propose a morphology randomization method that reduces the need for robot specific information during training by procedurally generating a diverse set of morphologies. By randomizing the morphology and dynamics of the simulated robots, our system is able to train generalized controllers that can be deployed across a variety of different robots, as shown in Fig. 1.

The core contribution of this work is the development of a reinforcement learning framework that produces generalized locomotion (GenLoco) controllers for quadrupedal locomotion skills, which can be deployed on a large variety of different robot morphologies. By training controllers on a wide range of randomly generated robot morphologies, our system is able to learn policies that generalize to new robots not seen during training. We demonstrate the effectiveness of our model on many notable quadrupedal robots in simulation and in real-world experiments. We show that the learned controllers can be deployed directly on a number of commonly used quadrupedal robots, including the Unitree's A1, MIT's Mini Cheetah [14] and CUHK's Sirius. We also demonstrate that, by introducing morphology randomization into the training process (varying robot parameters such as body size, leg length, and mass during training), our framework is able to more effectively transfer policies from simulation to the real world than policies trained on a specific robot. In this paper, we focus on the randomized robot morphology while keeping the number of Degree of Freedoms (DoFs) and links constant. The open-source GenLoco policies could be used as baselines to test controllers for newly developed quadrupedal robots without the need to train a robot-specific policy from scratch.

## 2   Related Work

RL provides a general framework for learning robotic controllers for a large array of tasks. Instead of requiring tedious manual controller engineering, RL techniques can automatically synthesize controllers for a desired task by optimizing the controller against an objective function [15]. RL has been applied to develop a wide range of motor skills for agents in simulation [16, 17, 18, 19], and in the real world [20, 21, 22, 23, 24, 5, 12]. But due to challenges associated with applying RL algorithms on real-world systems [25], sim-to-real techniques are commonly used to adapt and transfer controllers trained in simulation to a physical system [26, 27, 8, 7, 5, 28, 29, 13]. Domain randomization is one of the most commonly used sim-to-real transfer techniques, where the dynamics

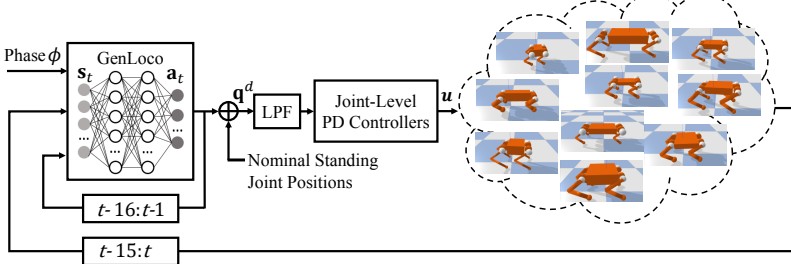

Figure 2: The proposed generalized locomotion control framework for quadrupedal robots. GenLoco is designed to work on a large collection of robots with different morphology and dynamics. The input observations of the policy consist of a phase variable $\phi$ representing progression along a motion, a history of the robot's raw sensor feedback, and a history of past actions. The actions output by the controllers are added to time-invariant nominal joint positions and passed through a low-pass filter before being applied to joint-level PD controllers to generate motor torques.

of a simulator are randomized during training in order to develop controllers that can adapt to differences between simulated and real-world dynamics [30, 31, 26, 8, 32, 33, 29, 34]. Real-world data can also be leveraged to further improve real-world performance by adapting the dynamics of the simulator [35, 36, 5, 37, 38], or directly finetuning a controller's behaviors [39, 7, 9, 13]. In this work, we also take a domain randomization approach to transfer controllers trained in simulation to robots operating in the real world. However, unlike many prior sim-to-real methods, which assume a known morphology for the real-world robot, our work explores the development of more general locomotion controllers deployable on a variety of different robots, including morphologies not known during training.

Developing general-purpose controllers that can be deployed on robots with different embodiments can greatly mitigate the overhead of creating individual robot-specific controllers. Devin et al. [40] utilized a modular network architecture to transfer manipulation skills across a small set of manually-crafted simulated robotic arms. Chen et al. [41] proposed conditioning policies on learned encodings of a robot's morphology, which allowed a controller to be deployed on a number of different robotic arms. A similar morphology encoding approach has also been applied to train full-body motion controllers, which can be deployed on a large variety of simulated humanoid agents [42]. Graph convolutional networks have been used to implicitly encode the morphological structure of simulated robots with varying numbers of degrees-of-freedom [43, 44]. Gupta et al. [45] learned domain invariant feature spaces in order to transfer manipulation skills across simulated robots with different morphologies. While these transfer learning techniques have shown promising results in simulation, they have yet to be demonstrated on robots in the real world. In this work, we aim to develop general locomotion controllers that can be deployed on a large variety of robots with different morphologies. Although prior works suggest that recurrent or memory-based policies can improve transfer by performing "implicit system identification" [26, 34], we found that our approach was able to produce effective policies with fully feedforward neural networks. By taking in past observations and actions as input, the model is able to to implicitly encode task-relevant information about the robot's morphology, and successfully transfer locomotion skills across a diverse set of quadrupedal morphologies without requiring an explicit representation of a particular robot's body structure.

## 3  Generalized Quadrupedal Locomotion Controllers

In this section, we present a framework for training a Generalized Locomotion Controller (GenLoco), where a single controller can be deployed on a large variety of quadrupedal robots with different morphologies. An overview of our system is shown in Fig. 2. The controllers are trained through model-free reinforcement learning. But unlike most prior RL frameworks for robotic locomotion, which assume a fixed morphology for the robot during both training and testing, our work explores the setting where the embodiment of the robot can vary, and the particular embodiment at test time is not known a priori. Our models are trained by randomizing the morphology and other dynamics properties during training in simulation, thereby encouraging the controller to learn adaptable strategies, which can be effectively applied on different robots. At every timestep, the controller receives a history of sensory observations as input, which can be used to infer task-relevant information about the robot it is currently deployed on. The controller then outputs actions, which are added to time-invariant

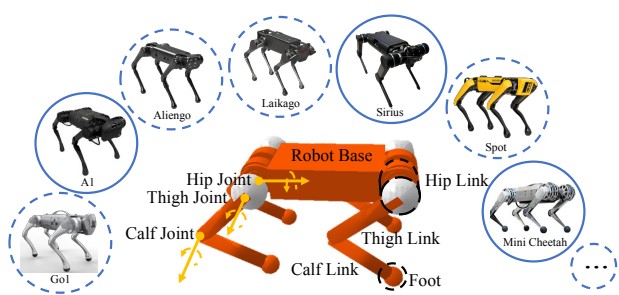

Figure 3: Many common quadrupedal robots follow a common morphological template, consisting of a robot base (6 DoFs) and four 3-DoF legs. This design is followed in robots such as Unitree's A1, Go1, Aliengo, Laikago, MIT's Mini Cheetah, CUHK's Sirius, and Boston Dynamics' Spot. The robots highlighted with solid circles are the ones used to validate our system in the real world.

Table 1: Morphology generation parameters. The nominal values are based on the A1 robot parameters (Appendix A).

| Parameter | Min and Max Values | Parameter | Min and Max Values |
|---|---|---|---|
| Size factor $\alpha$ | $[0.8, 1.2]$ | Calf length $l_c$ | $\alpha \times [0.11, 0.33]$ m |
| Base length $b_l$ | $\alpha \times [0.134, 0.400]$ m | Calf radius $r_c$ | $\alpha \times [0.01, 0.03]$ m |
| Base width $b_w$ | $\alpha \times [0.097, 0.291]$ m | Thigh length $l_t$ | $[0.75, 1.25] \times$ Calf length $l_c$ |
| Base height $b_h$ | $\alpha \times [0.057, 0.171]$ m | Thigh radius $r_t$ | $[0.75, 1.25] \times$ Calf radius $r_c$ |
| Base density $b_d$ | $[400, 1200]$ kg/m$^3$ | Foot radius $r_f$ | $1.5 \times$ Calf radius $r_c$ |
| Link masses | $[0.5, 1.5] \times$ nominal value | Joint PD gains | nominal value $\times \frac{\text{robot mass}}{\text{nominal mass}} \times [0.7, 1.3]$ |

nominal joint positions to specify target motor positions for each joint. The desired motor positions are passed through a low-pass filter to mitigate undesirable high-frequency movements [7], before being applied to joint-level PD controllers to generate motor torques.

## 3.1 Morphology Generation

The key to developing generalized locomotion controllers lies in training the controller on a diverse collection of robots, which encourages the controller to learn strategies that can generalize to new robots not observed during training. At the start of each training episode, we procedurally generate a random robot morphology based on a predefined morphology template. As illustrated in Fig. 3, many commonly used quadrupedal robots follow a similar body structure, consisting of a robot base and four legs, each of which has 3 DoFs: hip, thigh, and knee joints. The robot's feet can be modeled as spheres, and the contacts between the feet and the ground can be approximated as point contacts. This design template has been followed in widely used quadrupedal robots, including Unitree's A1, Go1, Laikago and Aliengo, Boston Dynamics' Spot, MIT's Mini Cheetah, and CUHK's Sirius. Therefore, our system generates random morphological variations based on this template by randomizing the proportions of the various body parts and their respective dynamics properties. A sample of robots produced by our morphology generation procedure are shown in Fig. 2. The randomized morphology parameters and their respective ranges are recorded in Table 1. Below, we provide a more detailed account of the major parameters of variations in our morphology generation procedure.

**Size factor.** We introduce a size factor $\alpha$, which uniformly scales the size of each body part (e.g. robot base and legs). Because $\alpha$ multiplicatively scales along every dimension, it is a sensitive parameter, and we found randomly sampling values of $\alpha$ from the range $[0.8, 1.2]$ to be sufficient in capturing the variation among most robots. This factor facilitates positive correlation between robot component sizes, reducing the likelihood of generating morphologies with distorted proportions (e.g., a robot with an extremely large base but extremely small legs).

**Robot base parameters.** The parameters of the robot base include its size and density. We model the base geometrically as a rectangular box. The dimensions, such as length $b_l$, width $b_w$, and height $b_h$, of the base are randomized to cover a wide range of robot base sizes. The density of the base is also randomized between $[400, 1200]$kg/m$^3$. The ranges of values, detailed in Table 1, are selected to encapsulate the sizes and masses of popular industrial robots as listed in Appendix E.

**Leg parameters.** The robot's upper leg (thigh) and the lower leg (calf) are modeled as cylindrical solids, and the robot's feet are represented by spheres, as shown in Fig. 3. Specifically, the sizes of the robot's thigh and foot are chosen based on the robot's calf link dimensions, as detailed in Table 1. The range of values for the thigh length $l_t$ is specified to be $\pm 25\%$ of the calf length $l_c$. Similarly, the range of values for the thigh radius is defined as $\pm 25\%$ of the calf radius $r_c$. By correlating the properties of different segments of the legs, we can prevent the morphology generator from producing implausible designs, such as a robot with large calves but very small thighs and feet, which would hinder the robot's locomotion capabilities.

**PD gains.** Larger and heavier robots typically require stiffer joint-level PD gains in order to generate larger motor torques. Therefore, the gains used in the PD controllers are also scaled with respect to the mass of the robot. The *nominal mass* is set to the mass of the A1 (12.458 kg). This randomization scheme helps to ensure that the PD controllers are sufficiently strong for larger and heavier morphologies.

## 4 Training

Our controllers are trained to perform various locomotion skills using a reinforcement learning-based motion imitation framework based on Peng et al. [7]. The goal of these controllers is to imitate a given reference motion $\mathbf{q}^r = \{\mathbf{q}_0^r, \mathbf{q}_1^r, \ldots, \mathbf{q}_T^r\}$, which specifies target poses $\mathbf{q}_t^r$ at each timestep $t$, The reward $r_t$ at timestep $t$ is computed according to:

$$r_t = w^{\mathrm{p}} r_t^{\mathrm{p}} + w^{\mathrm{v}} r_t^{\mathrm{v}} + w^{\mathrm{bp}} r_t^{\mathrm{bp}} + w^{\mathrm{bv}} r_t^{\mathrm{bv}},$$

$$\text{with } w^{\mathrm{p}} = 0.6, \quad w^{\mathrm{v}} = 0.1, \quad w^{\mathrm{bp}} = 0.15, \quad w^{\mathrm{bv}} = 0.15.$$

The pose reward $r_t^{\mathrm{p}} = \exp\left[-5\sum_{j=1}^{12} \|\hat{q}_t^j - q_t^j\|^2\right]$ encourages the robot to match its local joint rotations $\mathbf{q}_t$ with those specified by the reference motion, where $q_t^j$ denotes the rotation of the $j$-th joint. The velocity reward $r_t^{\mathrm{v}}$ follows a similar form, and encourages the robot to match the joint velocities of the reference motion. The base position reward $r_t^{\mathrm{bp}}$ and base velocity reward $r_t^{\mathrm{bv}}$ encourage the robot to track the motion of the base from the reference motion. A more detailed description of the reward function is available in Appendix B, and additional implementation details are described in Appendix C.

**State and action spaces.** As shown in Fig. 2, actions from the controller $\mathbf{a}_t$ specify target joint positions $\mathbf{q}^d \in \mathbb{R}^{12}$, which are used by the joint-level PD controllers to determine desired motor torques. The controller operates at 30 Hz. To encourage smoother motions, actions are processed with a low-pass filter before being applied to the robot. The observation $\mathbf{s}_t$ of the controller includes three components: 1) a 15-timestep history of robot raw sensor feedback $\mathbf{q}_{t-15:t}^{\mathrm{base}}$ and $\mathbf{q}_{t-15:t}$, 2) a 15-timestep history of past actions $\mathbf{a}_{t-16:t-1}$, and 3) a phase variable $\phi \in [0,1]$ is the normalized time that indicates the robot's progress along the reference motion, where 0 denotes the start of the motion and 1 denotes the end [16]. The robot's sensor feedback at each timestep consists of the base orientation $\mathbf{q}^{\mathrm{base}} \in \mathbb{R}^4$ from IMU sensors, recorded as a quotation, and the measured joint positions $\mathbf{q} \in \mathbb{R}^{12}$. Note that the observations do not include quantities that require an explicit state estimator, such as the linear velocity of the base [6, 7, 33, 11] or contact sensors [8]. The history of sensor readings and actions provides the GenLoco policy some information that can be used to perform state estimation and to infer the robot's dynamics and morphology.

**Dynamics randomization.** In addition to randomizing the morphological structure of the robots, we also randomize the dynamics parameters of the simulation during training [31] to improve the robustness of our policy and facilitate sim-to-real transfer. The randomized dynamics parameters include three categories: 1) link mass, link inertia, and ground friction to deal with modeling errors, 2) motor strength (torque limits), joint-level PD gains and motor damping ratio to mitigate the uncertainties of motor dynamics, and 3) the latency between the policy and the joint-level controllers. The randomization ranges of each parameter are detailed in Table 3 in Appendix D.

**Episode design.** Each episode has 100 timesteps, lasting about 3 seconds. Early termination is applied if the robot deviates too far from the reference base position and orientation [7]. At the start

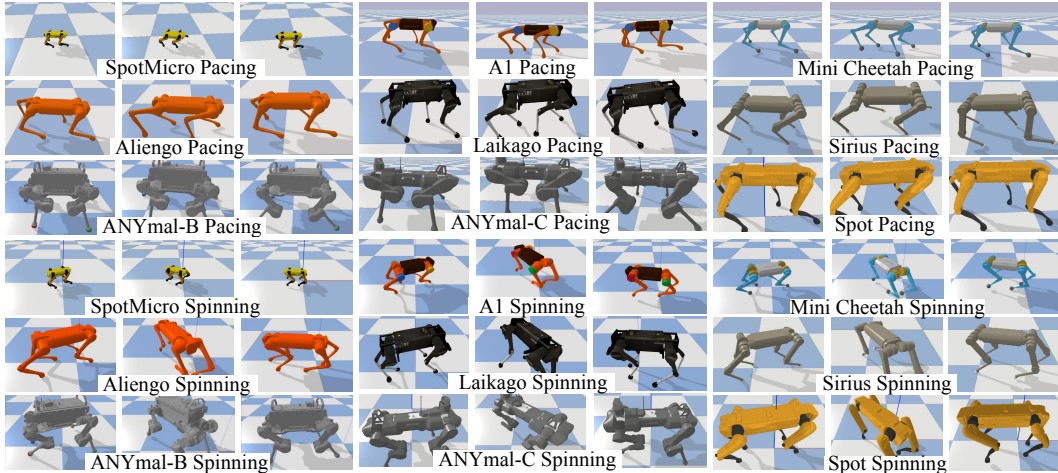

Figure 4: GenLoco policies deployed on a collection of quadrupedal robots that exist in real life. Two separate models are trained to perform a pacing gait and a spinning gait respectively. The GenLoco policies are trained in simulation using only procedurally generated robots, and robots used for testing are not used in the training process. The learned controllers can be directly deployed on all of these robots, including the ANYmal-series robots which have a distinct knee joint design, to perform agile maneuvers without additional training.

of each episode, a new morphology is generated according to the procedure described in Sec. 3.1, and the dynamics parameters are also randomized according to Sec. 4.

GenLoco policies $\pi_\theta$ are trained using Proximal Policy Optimization (PPO) [46], and all simulations are performed in Pybullet [47]. Each policy was trained with 800 million samples, taking approximately 2 weeks on 16 CPU workers.

## 5 Simulation Validation

In this section, we validate the effectiveness of our models on controlling quadrupedal robots with different morphologies by direct transfer in simulation. We compare our GenLoco policies with the policies trained on specific robots, which use the same training settings introduced in Sec. 4. We train controllers for imitating two reference motions: a forward moving pacing gait and an in-place spinning motion. The reference motions (using motion capture data from a real dog [7]) are retargeted to the A1's morphological features, and rescaled for each new morphology to account for differences in sizes and proportions. Thus, new reference motions are not required during test-time.

After being trained on procedurally generated morphologies, our controllers were tested on simulated models of commercially available robots using nominal standing poses and PD gains found in [48]. Robots used for testing include the A1, Aliengo, Go1, Mini Cheetah, Sirius, Laikago, Spot, SpotMicro, ANYmal-B, and ANYmal-C. The SpotMicro [49] is relatively tiny, with a mass of 4.8 kg and a fully-standing height of 0.26 m. The A1 and Mini Cheetah are small robots weighing about 10 kg and standing about 0.4 m in height. The Sirius, Aliengo, Laikago, and Spot are heavier and larger, each having height $\geq 0.5$ m. The morphological parameters of these robots generally lie within the training distributions. However, the ANYmal-B and ANYmal-C have a different knee joint design from the design template used for training. Detailed specifications of these robots are listed in Appendix E.

### 5.1 Zero-Shot Transfer to Novel Robots

As shown in Fig. 4 and the supplementary video, our controllers can be directly deployed on a large variety of quadrupedal robots. None of the robots we tested on were used during training. Notably, GenLoco policies successfully transfer to the ANYmal-series robots, which possess an inverted knee joint design different from the template used during training (Fig. 3), without additional tuning. This highlights the generalizability of our proposed GenLoco framework.

Our GenLoco policies are some of the first locomotion controllers that can be deployed directly on different quadrupedal robots without further fine-tuning. By providing a history of past observations and actions to the controller, we enable it to leverage the history to infer the morphology of the robot

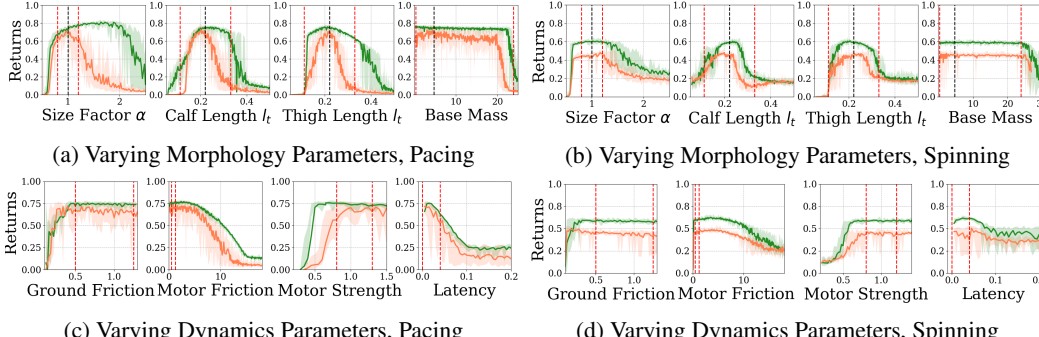

(a) Varying Morphology Parameters, Pacing      (b) Varying Morphology Parameters, Spinning

(c) Varying Dynamics Parameters, Pacing      (d) Varying Dynamics Parameters, Spinning

Figure 5: Benchmark of performance of GenLoco policies and policies trained specifically for A1 robot to perform pacing and spinning skills on a range of different robot morphologies and dynamics parameters in simulation. Green lines are the normalized return of the GenLoco policy (ours) while orange ones are those of the A1-specific policy. Returns are calculated by normalizing the cumulative reward (Appendix B) over the episode length. The red dashed lines indicate the training range while the black dashed lines denote the A1's morphological parameters. Note that for the varying morphology test the A1-specific policy is not trained with randomized morphology, and the dynamics randomization range is the same for all of these policies. Overall, the GenLoco policies outperform the A1-specific policies over different morphology and dynamics parameters. Furthermore, GenLoco is able to generalize over a larger range of morphologies and dynamics. Each testing episode lasts 100 timesteps and returns are averaged across 10 trials.

it is currently deployed on, and by randomizing the morphology during training the model develops more general control strategies that can effectively execute a desired skill on different morphologies.

## 5.2 Out-of-Distribution Generalization

To further evaluate the ability of GenLoco to generalize learned locomotion skills to different morphologies, we test the controllers extensively on a large range of simulated robot morphologies. We evaluate the controller's performance when varying four morphology parameters: 1) body size, which is scaled by size factor $\alpha$, 2) calf length $l_c$, 3) thigh length $l_t$, and 4) body mass $b_m$. The testing range is set to be much larger than the range used during training detailed in Table 1. We compare GenLoco to policies trained specifically for the A1, since the nominal values used in the morphology generation process (Table 1) are based on the original A1 robot. The comparisons with the pacing gait controllers and spinning gait controllers are shown in Fig. 5a and Fig. 5b, respectively. The GenLoco policies are overall able to better generalize to out-of-distribution robots, exhibiting more robust behaviors and maintaining higher returns across larger variations in the morphological parameters than the A1-specific policies. In the case of the pacing skill, GenLoco's performance does not degrade until the robot size is more than two times larger than the maximum training size. While the A1-specific policy shows some robustness to small variations in the morphology parameters, the policy fails when deployed on other robots, such as the Sirius.

In order to understand the advantages of the morphology randomization during training, we test the robustness of the GenLoco policies and A1-specific policies by varying the dynamics parameters. In addition to generalization to different morphologies, our model is also robust to large variations in the dynamics of a system. Fig. 5c compares the performance of GenLoco policies to A1-specific policies that were also trained using the same range of dynamics randomization detailed in Table 3. As demonstrated in Fig. 5c, 5d, for the same range of dynamics randomization during training, GenLoco policies consistently outperform the A1-specific policies. In particular, GenLoco policies demonstrate notable robustness to changes in the dynamics parameters related to actuated joints, such as motor friction and strength. This improved performance on out-of-distribution settings is likely in part due to random morphologies introducing a more diverse range of dynamics for the policy to train on.

## 6 Real-World Deployment

Finally, we evaluate the effectiveness of the learned controllers on robots in the real world. We test the controllers on three robots, Unitree's A1, MIT's Mini Cheetah and CUHK's Sirius. As shown in Fig. 6 and the supplementary video, our GenLoco policies can be deployed on the real robots in a

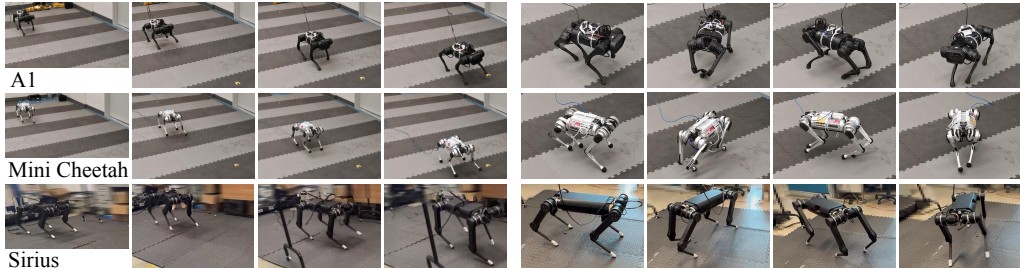

|   | (a) Pacing gait using a single GenLoco | (b) Spinning gait using a single GenLoco |
|---|---|---|

Figure 6: A single GenLoco policy can be deployed directly on different robots, which it was not trained on, in the real world, enabling them to perform agile maneuvers such as pacing and spinning.

Table 2: Normalized return (mean $\pm$ standard deviation) of policies deployed on real robots over 6 test repetitions

| *Pacing* | **GenLoco on A1** | **A1-Specific** | **GenLoco on Cheetah** | **Cheetah-Specific** |
|---|---|---|---|---|
| **Return** | $\mathbf{0.773 \pm 0.054}$ | $0.696 \pm 0.032$ | $\mathbf{0.743 \pm 0.092}$ | 0 (Failed in $6/6$ trials) |
| *Spinning* | **GenLoco on A1** | **A1-Specific** | **GenLoco on Cheetah** | **Cheetah-Specific** |
| **Return** | $\mathbf{0.670 \pm 0.070}$ | $0.572 \pm 0.049$ | $\mathbf{0.721 \pm 0.090}$ | $0.258 \pm 0.013$ |

zero-shot manner, enabling the robots to perform agile maneuvers such as pacing and spinning. We further compare the real-world performance of GenLoco to policies trained specifically for the A1 and Mini Cheetah. We performed 6 trials for each policy using the same experimental setup, and the performance of these policies are available in Table 2. We note that the performance of GenLoco policies, which work on both robots and were trained on neither, is overall better than the policies trained for specific robots, as shown in the supplementary video. Furthermore, the Mini Cheetah-specific pacing policy failed in all 6 real robot deployment trials, despite working well in simulation as shown in Fig. 4. Notably, such performance degradations during sim-to-real transfer do not occur for the GenLoco policies under identical environment circumstances. This suggests that solely randomizing the dynamics for a single robot may be insufficient due to differences in morphological characteristics between simulation and the real world. By training with randomized morphologies, the GenLoco policy better adapts to kinematics variations and therefore maintains robustness during sim-to-real transfer. An additional note of interest is that the timeline from completion of the Sirius hardware to our deployment of GenLoco control on Sirius was mere days. The successful zero-shot transfer observed in this experiment further highlights the utility of our model.

# 7 Limitations

As shown in Fig. 5a and 5b, the performance of GenLoco policies drops dramatically for larger robots. This may suggest that using more aggressive randomization and more expressive model architectures with recurrence, such as in [50], could improve the model. Moreover, our models cannot be deployed on robots with a different number of DoFs. More flexible architectures, such as graph neural networks [43, 44], may allow for adaptable models that can handle variable numbers of DoFs.

# 8 Conclusion

In this paper, we presented an RL-based framework for training generalized locomotion controllers that can be deployed on a diverse set of quadrupedal robots. We show that a simple history-based model, trained on procedurally generated robots in simulation, can be successfully transferred to a large variety of new robots, which were not observed during training. The trained models can also be deployed directly on real robots, without requiring any additional training on the physical systems. While our experiments have been focused on quadrupedal robots, our method is general and can also be applied to robots in other domains, such as robotic arms and quadrotors. However, the effectiveness of our models remains limited to robots that have the same number of DoFs and follow a predefined morphological template. Despite these limitations, we hope our work will provide a stepping stone towards more general-purpose controllers that can be widely and conveniently deployed on a diverse catalog of robots.

**Acknowledgments**

This work was supported in part by Hong Kong Centre for Logistics Robotics and in part by ARL DCIST CRA W911NF-17-2-0181. We thank Prof. Sangbae Kim, the MIT Biomimetic Robotics Lab, and NAVER LABS for lending the Mini Cheetah for experiments.

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
