# OpenReview forum: "GenLoco: Generalized Locomotion Controllers for Quadrupedal Robots"
_robot-learning.org/CoRL/2022/Conference — CoRL 2022 Poster_

### Official Review · Reviewer_3Unp · 2022-07-28

**Originality:** Good
**Technical Quality:** Very Good
**Clarity Of Presentation:** Very Good
**Impact:** 4

**Recommendation:**

Weak Accept: I recommend accepting the paper, but will not argue for my recommendation if the majority of other reviewers have a different opinion.

**Summary:**

This paper presents a reinforcement learning approach based on domain randomization for training unified adaptive locomotion controllers for generic quadruped robots with a fixed kinematic chain topology and a wide range of morphologies.

Given a target locomotion task, the approach trains a single policy in simulation by randomizing the quadruped’s links geometries and dynamics at each episode over a wide distribution encompassing those of many real-world quadruped designs.

A curriculum that linearly extends the parameter distribution support as training progresses is shown to speed up convergence.

Extensive experiments evaluate the learned controller on a wide range of quadrupeds in simulation and on two real-world ones on pacing and spinning gait locomotion tasks. It is shown to be more robust to varying parameters with respect to robot-specific policies and to be directly transferable to unseen real robots.


**Issues:**


Update: Authors' responses and improvements to the submitted material successfully address my concerns.

-------

Consider addressing the weaknesses listed above in the strengths and weaknesses section, i.e.,
- Experiment repetitions
- Single robot baseline fair comparison in real robot deployment experiment
- Successful deployment with inverted knee design discussion
- Better explain a few passages and concepts
- Other points listed above



**Quality Of The Limitations Section:**

Limitations are addressed clearly

**Reviewer Expertise:**

5: The reviewer is absolutely certain that the evaluation is correct and very familiar with the relevant literature

**Robotics Focus:**

Sufficient demonstration on hardware

**Strengths And Weaknesses:**

This work has the significant merit of empirically exploring the capabilities of relatively simple domain randomization and model-free RL techniques to a previously unseen range and scale in robot learning of complex locomotion tasks.
It demonstrates that a single policy trained with domain randomization over a high-dimensional distribution of morphologic and dynamic parameters allows to control a wide range of robots and, for the first time, can be directly applied beyond a single real robot with good performance.

From the technical point of view, the approach is relatively straightforward. It brings together several standard algorithms and methods for locomotion controller learning and domain randomization for sim to real transfer, Scaling them up well beyond previously published studies and demonstrating their effectiveness.

Experiments are generally well-executed and support generalization and transferability claims, although several statements would benefit from more analysis and in-depth discussion. I report my comments for each experimental section:

*5.1 Zero-Shot Transfer to Novel Robots*: The observation on successful deployment on ANYmal-B is purely descriptive and high-level. Too few details are provided on the setup and how joint references are adapted to work on such a different knee design with respect to the training distribution. This result is indeed interesting, but the reasons behind this successful transfer shall be more thoroughly discussed to be informative.
Also, at L213 authors claim that “we enable it to leverage the history to infer the morphology of the robot”. This appears too strong, since there are no detailed analyses of how the policy network actually employs the information included in the state and action historical data. It is very unlikely for it to infer the morphology of the robot explicitly. Still, one could say that the policy may learn to implicitly exploit part of the information contained in the history in a task-oriented way.

*5.2 Out-of-Distribution Generalization*: This section tests the robustness and generalization to out-of-distribution morphology and dynamics parameters of the proposed method. The plots reported in Figure 5 successfully supports the claim of better generalization of the proposed policy trained with domain randomization with respect to a baseline policy trained on a specific morphology. This enhanced generalization is quite expected by a policy learned on a wide distribution of morphologies.

*6. Real-world deployment experiments:* These experiments are relevant and successful in demonstrating the transferability of the learned adaptive controller to multiple different real robots. To my knowledge, this is the first individual locomotion policy successfully deployed on two different real-world quadrupeds, which represents a clear technical achievement.
Still, the comparison with robot-specific policies and the claims of superior performance are not fully convincing for several reasons. First, the number of repetitions of the comparison experiments in Table 2 is not reported, which does not allow to conclude on this matter. Secondly, it is not possible to conclude on the fairness of the single-robot baseline policy training. It is surprising that a successful locomotion policy targeted to a single robot (and thus to a simpler learning task) cannot be trained and its suboptimal behaviors cannot be dealt with by using standard sim-to-real approaches, e.g. by dynamics randomization. The paper does not report enough details on how the baseline policies are trained ad on the possible causes for suboptimal behaviors (i.e., drifting and failure).

The quality of presentation is generally high. The paper is well-structured, well-written and pleasant to read. Figures are high-quality and informative.

However, some parts of the method and results description appear to lack enough detail and depth. For example, some key elements like the policy network structure, initial state distribution (and reference state initialization?) are missing, and, as already mentioned above, some results and observed phenomena are only described without further analysis, e.g.: the reason why the policy may be compatible with a different knee design; the “morphology inference” enabled by historical data; reasons behind single-robot policies failures when deployed on real robots.

More comments on potential improvements:
- L24 Consider mentioning "Yang, Chuanyu, et al. "Multi-expert learning of adaptive legged locomotion." Science Robotics 5.49 (2020): eabb2174.", which represents a relevant line of work employing both mixture of expert models (inspired by PFNN and MANN) trained on reference data and RL and demonstrates real-world deployment on a quadruped.
- L62 Consider including the following relevant work, which to my knowledge is the first successful humanoid (not just biped) walking policy learned in simulation and successfully transferred to a real-world humanoid: Rodriguez, Diego, and Sven Behnke. *"DeepWalk: Omnidirectional bipedal gait by deep reinforcement learning." 2021 IEEE International Conference on Robotics and Automation (ICRA). IEEE, 2021.*
- L85 “Although prior works suggest that recurrent or memory-based policies can improve transfer by performing “implicit system identiﬁcation,” …”: consider explicitly citing relevant prior work
- L203 refers to ANYmal in Figure 3, where it is not actually shown
- Minimum base length, width and height in Table 1 look way too small
- L123 “This factor facilitates positive correlation between robot component sizes”: unclear statement
- L144 “Our controllers are trained to perform various locomotion skills using a reinforcement learning-based motion imitation framework [3]”: The proposed approach does not seem to use exactly the same motion imitation framework proposed in [3], that learns a latent space representing morphological variability. Which parts are included and which ones are not?
- What is the initial state distribution? Is reference state initialization employed?

Others:
- L52 real-world $/rightarrow$ real world
- L80 trained $/rightarrow$ learned
- L88  to infer the morphological structure of a robot $/rightarrow$ to implicitly encode task-relevant information about the robot’s morphology (?)



**Summary Of Recommendation:**

Overall, the paper presents relevant results showing the successful scaling of existing domain randomization techniques for RL to unprecedented levels. Also, the first successful deployment of a single learned adaptive locomotion controller on two different quadruped robots in the real world represents a technological breakthrough.

Authors' responses and improvements to the submitted material successfully address my concerns.

---

> ### Author Response · Authors · 2022-08-22
> **Response to Reviewer 3Unp**
>
> Thank you for your detailed suggestions and comments. We have incorporated them into our revision (attached under our General Response) and below are more in-depth responses to your questions:
>
> ---
>
> > The observation on successful deployment on ANYmal-B is purely descriptive and high-level. Too few details are provided on the setup and how joint references are adapted to work on such a different knee design with respect to the training distribution.
>
> Thanks for the pointing this out. Please refer to the general response (section "Insight on why GenLoco can transfer to ANYmal-series robots") for a detailed answer.
>
> ---
>
> > at L213 authors claim that “we enable it to leverage the history to infer the morphology of the robot”. This appears too strong, since there are no detailed analyses of how the policy network actually employs the information included in the state and action historical data. It is very unlikely for it to infer the morphology of the robot explicitly. Still, one could say that the policy may learn to implicitly exploit part of the information contained in the history in a task-oriented way.
>
> > L88 to infer the morphological structure of a robot $/rightarrow$ to implicitly encode task-relevant information about the robot’s morphology (?)
>
> Thanks for this suggestion. We agree that our original tone may be too strong, and we have updated the paper with the suggested passage.
>
> ---
>
> > the comparison with robot-specific policies and the claims of superior performance are not fully convincing
>
> > The paper does not report enough details on how the baseline policies are trained ad on the possible causes for suboptimal behaviors
>
> > reasons behind single-robot policy failures when deployed on real robots
>
> Thanks for this question, please refer to the general response (sections "Explanation on why GenLoco outperforms robot-specific policies," "Hardware experiment reproducibility," and "Fairness of robot-specific policy compared with GenLoco policies") for a detailed answer. We also include a discussion in the updated "Real-World Deployment" experiments section of the revision.
>
> ---
>
> > some key elements like the policy network structure, initial state distribution (and reference state initialization?) are missing
>
> Thanks for pointing this out. We represent the actor policy network using an MLP with hidden layers [1024, 512] and ReLU nonlinearity. The network for the value function uses the same structure for the hidden layers and uses linear activation for the final output layer. During the initialization of each episode, the robot uniformly samples a pose from the reference motion as its initial pose with 90% probability, and is initialized from a nominal standing pose with 10% probability.
>
> We have added these details in Appendix C of the revision.
>
> ---
>
> > Minimum base length, width and height in Table 1 look way too small
>
> Thanks for pointing this out, there was a copy-paste issue from the code to the paper. It has been fixed in the revision.
>
> ---
>
> > L123 “This factor facilitates positive correlation between robot component sizes”: unclear statement
>
> We use a scaling factor that is multiplicatively applied to all the different parameters of a generated morphology; this causes the different dimensions of the robot’s base and legs to be correlated (since they are all multiplied by the same randomized scaling value $\alpha$). This effectively reduces the probability of infeasible ratios between robot morphological component sizes, such as the case where the base is extremely large but the legs are extremely small.
>
> We have added this clarification in the revision.
>
> ---
>
> > L144 “Our controllers are trained to perform various locomotion skills using a reinforcement learning-based motion imitation framework [3]”: The proposed approach does not seem to use exactly the same motion imitation framework proposed in [3], that learns a latent space representing morphological variability. Which parts are included and which ones are not?
>
> The latent space learned in *Learning Agile Robotic Locomotion Skills by Imitating Animals* [3] represents dynamics variability rather than morphological variability. Our framework is built on the simulation environment in PyBullet provided in [3], but with the addition of randomly generated robotic morphologies. We do not use the latent variable adaptation module from [3], instead using a simple MLP policy which we zero-shot transfer to the hardware.
>
> We have clarified this in the "Training" methodology section of the revision.
>
> ---
>
> Finally, thanks for the suggested additional references, we have added them to the paper.

---

> > ### Comment · Reviewer_3Unp · 2022-08-27
> > **Acknowledgment of Paper394 Authors' Response**
> >
> > I would like to thank the authors for their detailed responses and updates to the main text, which cover most of my questions and remarks.
> >
> > The additional discussions and experiment repetitions details about GenLoco's performance with respect to specific policies in the sim-to-real experiments seem satisfactory and strengthen the paper.
> >
> > Additional successful experiments on larger robots like Spot (in simulation) and Sirius2 (real world) also contribute to demonstrate the generalization properties of the GenLoco-trained policy.
> >
> > I acknowledge the removal of the curriculum learning statemets, based on comments made by other reviewers. In my view, this does not negatively impact the overall contribution.
> >
> > Pending potential discussions with the other reviewers and AC, I am now oriented towards suggesting acceptance.

---

### Official Review · Reviewer_GntH · 2022-07-28

**Originality:** Excellent
**Technical Quality:** Very Good
**Clarity Of Presentation:** Good
**Impact:** 4

**Recommendation:**

Strong Accept: I recommend accepting the paper and will argue for my recommendation even if other reviewers hold a different opinion.

**Summary:**

The paper proposed Generalized Locomotion Controller (GenLoco) which can be deployed on a wide variety of quadrupedal robots to produce locomotion behaviors.
The main contributions of the paper are following two points:
1. The authors developed a morphology randomization method to learn a generalized policy of GenLoco.
2. The authors showed the network architecture and learning framework to design GenLoco.


**Issues:**

* I am curious whether the randomization degrades the best control performances or not. I would like to see the comparisons between the GenLoco policies and non-randomization policies (A1-specific policies without the dynamics randomization).
* I would like to know the practical situations, applications or scenarios where the advantage of mitigating the overhead of creating individual robot-specific controllers is beneficial. GenLoco should also be evaluated in such or similar situations.


**Quality Of The Limitations Section:**

Limitations are addressed clearly

**Reviewer Expertise:**

3: The reviewer is fairly confident that the evaluation is correct

**Robotics Focus:**

Sufficient demonstration on hardware

**Strengths And Weaknesses:**

Strength:
* The authors successfully demonstrated that the locomotion behaviors of a large collection of simulated quadrupedal robots could be generated by GenLoco.
* The authors successfully demonstrated that the GenLoco policy, trained on the simulated robots, could be transferred to real quadrupedal robots A1 and Mini Cheetah. Surprisingly, the control performances of the robot-non-specific policy (GenLoco) are superior to those of the robot-specific policy. The potential impact of the paper is high.

Weakness:
* Although the authors mentioned that GenLoco could reduce the labor-intensive process of developing robot-specific controllers for new systems, it is difficult for me to imagine when such advantages are beneficial. The practical situations, applications or scenarios should be discussed, and GenLoco should be evaluated in such or similar situations. For example, in what kind of situations must robots with similar but different morphologies work together?

Minor comments:
* A typo: Our Models in Line 98.


**Summary Of Recommendation:**

The contribution of this paper is sufficient: strength in both originalities and potential impact of the approach. The experiments are performed in not only simulation but also real environments.

---

> ### Author Response · Authors · 2022-08-22
> **Response to Reviewer GntH**
>
> Thank you for the comments and feedback.
>
> > I am curious whether the randomization degrades the best control performances or not. I would like to see the comparisons between the GenLoco policies and non-randomization policies (A1-specific policies without the dynamics randomization).
>
> Thanks for the question. The role of dynamics randomization for quadrupedal robots has been previously studied in detail in Peng et al. [3], where a quadruped trained without dynamics randomization failed in deployment on the physical robot while training with dynamics randomization yielded improved performance. Please see timestamp 4:42 (no rand vs. robust) of the video (https://youtu.be/lKYh6uuCwRY?t=282) accompanying the Peng et al. paper [3]. We therefore chose not to re-include this ablation test in our work.
>
> > I would like to know the practical situations, applications or scenarios where the advantage of mitigating the overhead of creating individual robot-specific controllers is beneficial. GenLoco should also be evaluated in such or similar situations.
>
> One advantage of having a general locomotion controller is that it can remove the need to re-train new controllers for every robot, which can be time and compute intensive. We plan to release our trained controllers, so that others in the community can directly use them as baselines to test and control newly-developed robots, without the need to train new controllers from scratch each time. Therefore, GenLoco policies are tested on a large range of different virtual robots, including out-of-training-distribution ones as shown in Fig. 5, in the hope that some of them may be built in real life someday and we can directly deploy a GenLoco policy as a baseline controller for them. We have updated the contribution section of the introduction to emphasize this.

---

### Official Review · Reviewer_62wF · 2022-07-30

**Originality:** Very Good
**Technical Quality:** Very Good
**Clarity Of Presentation:** Very Good
**Impact:** 4

**Recommendation:**

Strong Accept: I recommend accepting the paper and will argue for my recommendation even if other reviewers hold a different opinion.

**Summary:**

This paper proposes a method to train generalized controllers for quadrupedal robots. The generalization in this paper means that a single controller can be deployed on various quadruped robots having similar morphologies such as A1, Mini Cheetah, or Laikago. The key idea is a parameterization of simulated quadruped robots, where the parameters include physical properties (e.g., shape, inertia, joint frictions, etc) and its ranges are designed to cover the entire spectrum of the target quadruped robots. A locomotion controller is then learned by deep reinforcement learning, where a simulated character is randomly instantiated every episode with domain randomization.

**Issues:**

Please see above.

**Quality Of The Limitations Section:**

Additional details required

**Reviewer Expertise:**

4: The reviewer is confident but not absolutely certain that the evaluation is correct

**Robotics Focus:**

Sufficient demonstration on hardware

**Strengths And Weaknesses:**

Strengths

- The formulation is simple and straightforward, but it works pretty well not only in the simulation but also in the real world, which is impressive.

- The controller does not require any state predictions or explicit parameters for physical properties of the robot, which makes hardware deployment easy.

- The learned algorithm was tested for the two real robots and it worked without any special treatment.

Weaknesses

- A learned controller can only perform a single behavior. Different behavior requires a different controller.

- The claim on the benefit of the proposed curriculum learning might be wrong (please see below for the details)


Suggestions for Improvement

- The controller learned with morphology randomization showed better performance in general over the specialized controller for the A1 robot (Figure5). The most surprising/impressive part here for me was that this benefit is even true when testing with the A1 robot in the simulation only. Did morphology randomization help the Deep RL algorithm to find a better solution by preventing it from converging to suboptimal solution in the early learning phase? If this claim can be verified, the paper could become more impactful. I really want to see more discussion on this in the revised version.

- Details for my comment above “The claim on the benefit of the proposed curriculum learning might be wrong (please see below for the details)”: The performance (returns) gap between the learning with/without curriculum in Figure7 is the main evaluation result supporting the claim. First, I don’t think that the initial gap is not meaningful because the challenging morphologies are not presented for the learning with curriculum. Second, there is nothing in the gap when looking at the gap around 200M where the two learning algorithms see the same morphology random distribution (the explanation in the paper: “The morphology randomization curriculum linearly anneals the range of parameter values from 0 to their respective maximum range over the course of 200 million samples”). This actually means that the curriculum is not helpful. Finally, it was mentioned that the authors collected 800M tuples for their final results, and the green curve (the algorithm without curriculum) is still increasing. So, the figure should be shown upto 800M.

- In the state representation, why is the time window different for the base joint (t-15:t) and the remaining joints (t-15:t-1)?

- How was the history of q_base, q, and actions set when t=0?

- “The reference motions are retargeted to match our robot’s morphology” in Line197 -> Do you perform retargeting every episode whenever a quadruped robot is randomly generated?

- Figure5 : It might be good to have a vertical line that matches the A1 robot.

- It is unclear how the ANYmal-B could be directly controlled by the proposed method. Isn’t the kinematics tree structure a bit different from what the parameterized controller can cover? I guess there were some technical tricks for setting the zero pose. If it exists, please provide more details.


**Summary Of Recommendation:**

The paper presented that a controller learned in the simulation with morphology and domain randomization can directly be deployed in the real world, it could even be helpful in cases where we only develop a controller for a specific robot. I think this factor is impactful enough and could be a stepping stone for future research.

---

> ### Author Response · Authors · 2022-08-22
> **Response to Reviewer 62wF**
>
> Thank you for the comments and the improvements that you’ve suggested. We address your questions below:
>
> > Did morphology randomization help the Deep RL algorithm to find a better solution by preventing it from converging to suboptimal solution in the early learning phase? If this claim can be verified, the paper could become more impactful. I really want to see more discussion on this in the revised version.
>
> Thanks for the question. We have addressed this in the general response section.
>
> > The claim on the benefit of the proposed curriculum learning might be wrong
>
> Thanks for pointing this out. We recognize that using a curriculum doesn’t make a large enough difference when given enough training samples, and therefore this section has been removed from the Appendix.
>
> > In the state representation, why is the time window different for the base joint (t-15:t) and the remaining joints (t-15:t-1)?
>
> This is a typo and should be $(t-15:t)$ for both the base joint and the other joints. It has been fixed in the updated version.
>
> > How was the history of q_base, q, and actions set when t=0?
>
> To initialize the history of the observations, $\mathbf{q}^\text{base}$ and $\mathbf{q}$ use the current measured value repeated $15$ times, while past actions are all initialized as zeros.
>
> > “The reference motions are retargeted to match our robot’s morphology” in Line197 -> Do you perform retargeting every episode whenever a quadruped robot is randomly generated?
>
> We retargeted motions once (during training) to the A1’s morphological features, and then for each new morphology we rescale the reference motion to account for differences in sizes and proportions. Please note that we do not need a reference motion for testing new robots, as it is not required by the observations in the policy. We have made this point more clear in the paper.
>
> > Figure5 : It might be good to have a vertical line that matches the A1 robot.
>
> Thanks for the suggestion, we have added it accordingly in the revised version of the paper.
>
> > It is unclear how the ANYmal-B could be directly controlled by the proposed method. Isn’t the kinematics tree structure a bit different from what the parameterized controller can cover? I guess there were some technical tricks for setting the zero pose. If it exists, please provide more details.
>
> Thanks for the question. We have addressed it in the general response section.

---

> > ### Comment · Reviewer_62wF · 2022-08-24
> > **Thanks for your response.**
> >
> > Thanks for your response. The rebuttal answers my questions and clarifies some missing details. I would stand for the original rating "strong accept".

---

### Official Review · Reviewer_KDQ2 · 2022-08-01

**Originality:** Good
**Technical Quality:** Good
**Clarity Of Presentation:** Excellent
**Impact:** 3

**Recommendation:**

Weak Accept: I recommend accepting the paper, but will not argue for my recommendation if the majority of other reviewers have a different opinion.

**Summary:**

The paper focuses on learning generalized locomotion policies that can be applied to a variety of quadrupedal robots of similar morphology (num DOFs and links are constant). The paper procedurally generates robots with randomized morphology and dynamics and then trains locomotion policies on these robots via a morphology curriculum.

The paper states that policy transfer across morphologies has been achieved before but only in sim.

The paper shows transfer to multiple robots in simulation including the AnyMal-B without fine tuning and shows that morphology randomization allows better OOD performance. The paper also shows transfer to two robots in the real world and shows that morphology randomization produces higher return.

In my opinion the primary contribution of the paper is showing that simple randomization strategies and policy networks can enable learning of policies that are applicable across multiple robots in simulation and the real world.


**Issues:**

Please address the questions in the Weaknesses section earlier.

Here are some additional minor comments----

Robot’s feet are modeled as spheres, and contacts are approximated as point contacts. How do we know these are sufficient approximations?

Why 15 states/actions? How was that number chosen?

Refs 1 and 19 are repeated

“Code and obtained policies”: what will the code entail exactly? How will it be useful to others?
What orientation representation is used? (I assume q indicates quaternion) What difference metric is used? (I assume this is the standard quaternion difference)

Line 159: q_base is 16 time steps, and q is 15 time steps. Is this intentional?

Please explain how the phase variable is determined. Is it just normalized time?

Line 164: “explicitly” → “explicit”

Appendix C: motor friction is in units of Nm? So it’s friction torque and not a coefficient?

Appendix C: what is “motor strength”? What does this correspond to in the real world and what are the units of this parameter?

Lines 175-176: episode length and horizon length are the same?

Appendix D: Units are millions of samples, or hundreds of millions of samples?

At test time are the joint PD gains equal to (nominal value * robot mass / nominal mass)?

Figure 5: The red lines in the top row are only applicable to the GenLoco policies, right, since A1 didn’t have morphology randomization? If so you might add another vertical line there to show what the specific A1 parameters were. It could also be really useful to add vertical lines or markers to show what the parameters were for the other robots on these plots. You have some remaining space in the paper and you could make these plots bigger.

Video: It would be good to visualize the reference trajectories alongside the real robots, otherwise it’s hard to know whether it’s doing the right thing.

Video: The drifting behavior isn’t too clear to me. Consider showing a better video of that.

Video: What are the white boxes that appear?

**Quality Of The Limitations Section:**

Limitations are addressed clearly

**Reviewer Expertise:**

4: The reviewer is confident but not absolutely certain that the evaluation is correct

**Robotics Focus:**

Sufficient demonstration on hardware

**Strengths And Weaknesses:**

----Strengths

The paper is well written, well illustrated and easy to follow.

The paper shows transfer to multiple robots in simulation including the AnyMal-B without fine tuning and shows that morphology randomization allows better out of distribution performance. The paper also shows transfer to two robots in the real world and shows that morphology randomization produces higher return.

Figure 5 nicely shows out of distribution results but could be better illustrated and labeled (see suggestions later in review).

Lines 239-240 may be the most important sentences in the paper. Maybe there’s a general lesson to be learned here that even with strong kinematics models, dynamics randomization is not enough for robust performance in the real world, and kinematic randomization should also be performed. You may want to expand on this a bit more.

----Weaknesses and questions

The paper’s methodology borrows very heavily from existing works, namely DeepMimic and Learning Agile Robotic Locomotion Skills, including identical or nearly identical actions, controllers, reward formulations, reference data and simulator.

Some randomization values seem quite specific and the paper doesn’t provide much insight as to how one should in general determine these randomization ranges. One thing that would help is a table of parameter values for the real robots so that they can be compared to the randomization ranges used in the paper. [Edit: I see that some real world values are provided in Appendix E, but several are still missing.] It would also be good to have a more specific written justification for how the ranges were chosen.

Link masses are not discussed in the explanation of parameters. Also, what is the “nominal value” and how was it chosen? [Edit: This seems to be given in Appendix A. This should be referenced in caption of Table 1 or in the description of table parameters.]

What is the “nominal value” for the PD gains? How was it chosen?

The reward function weights seem arbitrarily determined or borrowed from previous literature. How should a researcher determine these weights for the first time? How sensitive is policy learning to variations in those weights?

Only two reference motions were used, which seems quite limited. More reference motions should be examined in order to strengthen the paper.

Appendix D: Are the learning curves for one seed? If so, they should be shown for multiple seeds. Why does curriculum learning performance degrade? It looks like no-curriculum might even surpass curriculum if the graph is extended further. Longer term performance should be shown. The evidence for the benefit of curriculum learning does not seem convincing enough.

Why did transfer to AnyMal B work despite the very different morphology? What was the intuition?

As I understand, section 5.1 shows how training on procedurally generated robots transfers to real robot morphologies in simulation. I think the paper could be clearer that although the agent didn’t train specifically on those real robot morphologies, those morphologies were within the domain randomization distribution. If they weren’t, please state that clearly

Figure 5: the out of distribution performance seems impressive. However I’m not exactly clear why GenLoco would perform better than an A1-specific policy at the point where the test time dynamics parameters exactly equaled the actual A1 dynamics parameters. Can you provide some insight here?

Figure 5: What does return actually mean here on an absolute level? It would be good to know what this actually means in terms of for example mean joint position error, mean joint velocity error, and mean body pose error.

I don’t believe RLOC (Gangapurwala et al) was cited in this paper and I believe it should be compared to as well.


**Summary Of Recommendation:**

In terms of methods the paper borrows very heavily from existing works, namely DeepMimic and Learning Agile Robotic Locomotion Skills, including identical or nearly identical actions, controllers, reward formulations, reference data and simulator. It might be the case that there is a new approach proposed in combining [morphology and dynamics randomization] and [using history of states/actions with feedforward networks]. However I am not confident about that and would like the authors to verify if that’s accurate. Also a nice array of robots were examined (including 2 in the real world) but only 2 reference motions were evaluated.

In terms of results there seems to be a new result in showing that morphology and dynamics randomization can achieve reasonable out of distribution performance in simulation and importantly can achieve and improve sim to real transfer. This might point to a lesson that to achieve robustness in the real world, morphological randomization can be useful even if there are strong priors on the robot kinematic model. The authors should consider expanding on this.

Therefore I feel there’s maybe an incremental methodological advance but a useful and practical result. The effort to validate on multiple real quadrupeds is also commendable.

---

> ### Author Response · Authors · 2022-08-22
> **Response to Reviewer KDQ2**
>
> Thank you for your comments and feedback. We have addressed and corrected the fixes you suggested in our revision (attached under our General Response), and hope that the following responses address your concerns:
>
> ---
> > What is the “nominal value” for the PD gains? How was it chosen?
>
> > At test time are the joint PD gains equal to (nominal value * robot mass / nominal mass)?
>
> The nominal value of PD gains, which we have added to Appendix A, are based on the A1. During training, the PD gains are scaled by the robot size because heavier robots may need larger PD gains.
>
> PD gains are inherent attributes of each robot itself, so we did not use a pre-computed formula to determine test-time PD gains. Rather, during testing, these were specified independently as parameters of each existing robot which we found in the robot's open-source online URDF / codebase.
>
> ---
> > The reward function weights seem arbitrarily determined or borrowed from previous literature. How should a researcher determine these weights for the first time? How sensitive is policy learning to variations in those weights?
>
> Reward engineering is a tedious component of most reinforcement learning systems. Reward tuning is not a focus or contribution of this work, and we have largely just followed the reward function used for a single robot in Peng et al [3]. Note that we did not tune the reward when we were randomizing the robot’s morphology during training.
>
> ---
> > The evidence for the benefit of curriculum learning does not seem convincing enough.
>
> Thanks for pointing this out. We recognize that using a curriculum doesn’t make a large enough difference when given enough training samples, and therefore this section has been removed from the Appendix.
>
> ---
> > Why did transfer to AnyMal B work despite the very different morphology? What was the intuition?
>
> Thanks for the question. We address it in the general response.
>
> ---
> > As I understand, section 5.1 shows how training on procedurally generated robots transfers to real robot morphologies in simulation. I think the paper could be clearer that although the agent didn’t train specifically on those real robot morphologies, those morphologies were within the domain randomization distribution. If they weren’t, please state that clearly
>
> The existing robots that we tested in simulation in Fig. 4 are mostly within the morphology randomization distribution. However, we also tested the GenLoco policies on out-of-distribution “virtual” robots, as recorded in Fig. 5, and on existing robots which are not within distribution such as ANYmal-B and ANYmal-C, to demonstrate generalizability. We have clarified this point in the revision.
>
> ---
> > Figure 5: the out of distribution performance seems impressive. However I’m not exactly clear why GenLoco would perform better than an A1-specific policy at the point where the test time dynamics parameters exactly equaled the actual A1 dynamics parameters. Can you provide some insight here?
>
> Thanks for the question. We address it in the general response section.
>
> ---
> > Figure 5: What does return actually mean here on an absolute level? It would be good to know what this actually means in terms of for example mean joint position error, mean joint velocity error, and mean body pose error.
>
> The return is the cumulative reward over a 100-timestep episode, where the reward function computed for each timestep $t$ is detailed in Appendix B. In Fig. 5, the mean normalized return was calculated over 10 testing episodes in simulation (we have updated the caption to clarify this).
>
> ---
> > Robot’s feet are modeled as spheres, and contacts are approximated as point contacts. How do we know these are sufficient approximations?
>
> Most modern designs of quadrupedal robots have spherical toes, and given that our models correctly control the real robots, using spherical geometry seems to suffice. However, foot geometry indeed can impact performance, so to further improve generalizability one could also randomize foot geometry during training.
>
> ---
> > “Code and obtained policies”: what will the code entail exactly? How will it be useful to others? What orientation representation is used? (I assume q indicates quaternion) What difference metric is used? (I assume this is the standard quaternion difference)
>
> We will release our code for GenLoco training and testing as well as the models. We represented orientation with quaternion, and the orientation error calculation in the reward uses the standard quaternion difference metric.
>
> ---
> > Please explain how the phase variable is determined. Is it just normalized time?
>
> Yes, it is normalized time w.r.t. the reference motion timespan, where 0 denotes the start of the reference motion and 1 denotes the end. We have clarified this in the revision.
>
> ---
> > Appendix C: what is “motor strength”? What does this correspond to in the real world and what are the units of this parameter?
>
> "Motor strength" represents the motor's torque output limits (units N·m).

---

### Author Response · Authors · 2022-08-22
**General Responses (with Revised Paper/Video attached)**

**Comment:**

We would like to thank all reviewers for their insight and feedback. We have attached updated versions of the paper (changes highlighted in blue) and video to this comment. Below, we address common points raised in the reviews, and more specific responses will be provided to each reviewer through individual comments.

### Transfer to large robots
Further experiments that we recently ran also indicated that GenLoco policies can transfer to big robots, such as Spot and ANYmal-C, after correcting robot joint orders. The results are updated in Fig. 4 and the video attached. We plan on adding a discussion of common engineering errors when deploying GenLoco to different robots.

### Clarification on action space
We missed a part in the control diagram in Fig. 2. The action (output from the policy) is added to time-invariant nominal standing joint positions, which represent the robot default pose when it is standing, before applying the target positions to the joint-level PD controllers. During testing and deployment on a robot, we need to provide nominal standing joint positions of the robot. The nominal standing pose is not a part of the observation. During training, this nominal pose is fixed as the A1’s standing pose. We have updated the Fig. 2 diagram and action space definition.

### Hardware experiment reproducibility
The real-world experiments are repeated 6 times for each model, and the results remain consistent, as recorded in the updated Table 2 and the attached video.

### Fairness of robot-specific policy compared with GenLoco policies
The training setting (except the morphological randomization introduced in Sec. 3.1) and test-time experiment setting (including the environment and code) of robot-specific policies are identical to the GenLoco policies. Each robot-specific policy is trained to convergence using 200M samples.

### Explanation on why GenLoco outperforms robot-specific policies
We believe the two primary reasons are:
* Combining morphology randomization and dynamics randomization broaden the diversity of data used to train a policy, which can help the policy to better generalize the learned locomotion skill to the robots used for testing. This could be one of the reasons why GenLoco policies outperform robot-specific policies in the varying dynamics test in Fig. 5(c)(d), even though they are trained using the same range of dynamics randomization.
* During the sim-to-real transfer, the differences between the robot hardware and simulation include not only dynamics parameters but also kinematics and morphological characteristics. By training with randomized morphologies, the GenLoco policies can better adapt to variations in the kinematics, and therefore, generalize better during the sim2real transfer, as demonstrated in the experiments.

A more general takeaway from this paper is that, even when we focus on training a robot-specific policy for legged robots, randomizing dynamics parameters may not be enough, and adding morphology randomization could be helpful to improve the robustness of the policy. We have detailed this in the paper.

### Insight on why GenLoco can transfer to ANYmal-series robots (which share the inverted knee design that differs from the training template)
We believe there are two primary reasons:
* Although the knee joints of the ANYmal-series robots are inverted (w.r.t the morphological template we used in training), the resulting gaits on ANYmals are also “inverted” compared to the pacing gait on A1-type. Since the policy is robust to different quadrupedal dynamics, and we don’t explicitly provide the reference motion in the observation, even if the gait is mirrored compared to the original pacing gait (as in the ANYmal-series robots) the policy can still transfer.
* Parameterizing the actions as deltas on a nominal standing pose helps to mitigate some of the differences due to the inverted knee joints.

### Computation time
Although the GenLoco policy took about 800 million samples in total to train to convergence (which required 2-weeks of training on a 16-core CPU) compared to 200 million for each robot-specific policy, GenLoco has been successfully deployed on 8 different robots in simulation, which would have taken in total 1600 million samples to train separately. The GenLoco policy is also able to generalize to different other robots while robot-specific policies are unable to. Furthermore, utilizing more CPUs or moving the simulation environments to Issac Gym with GPU acceleration could help to drastically alleviate the training time.

**Zip File:**

/attachment/4f36f236a4d32e63762142e4d0a6e66a0d948fb2.zip

---

> ### Author Response · Authors · 2022-08-27
> **Recent Real-world Experiments on a Larger Quadrupedal Robot (with Video attached)**
>
> **Comment:**
>
> We recently deployed the GenLoco policies on a larger quadrupedal robot, Sirius 2, and it can also successfully be transferred to the real world. This robot has a mass of 23.7 kg and a height of 0.54m when fully standing. Please note that the physical robot was completed only days ago and our GenLoco policies can already be directly deployed on it. These experiments highlight the utility of GenLoco. Our policies can be used directly to allow newly developed quadrupedal robot to perform useful behaviors. Users can directly use GenLoco policies as baseline control policies to test their hardware without building or training new controllers from scratch.
>
> **Zip File:**
>
> /attachment/9e5a83566a299945d7910c0166065d409055072c.zip

---

### Meta-Review · Area_Chair_nbyj · 2022-08-04

**Recommendation:** Accept (Poster)
**Confidence:** 4

**Metareview:**

Please check the comments of the reviewers in detail.

### Strengths
- interesting and promising results
- relatively simple method
- hardware experiments on 2 different quadruped robots

### Weaknesses
- analyses could be deeper (see the reviewers' comments)
- incremental advance: randomization is not a novel idea and the paper almost reads like an in-depth "demonstration" of what is possible with current algorithms and computing power
- computing time: each policy is trained with about 800 million samples, requiring approximately 2 weeks on 16 CPU workers.
- statistical analysis is unclear: how many replicates for each experiment?
- did not work for bigger robots (see limitations)

### Post-rebuttal update
I would like to thank the authors for their efforts to improve the submitted paper. While some limitations remains (in particular, too few replicates for strong statistics and a large computation cost), the reviewers are mostly happy with the paper.